# NMR spectroscopy reveals acetylsalicylic acid metabolites in the human urine for drug compliance monitoring

**Galina Kupriyanova, Vladimir Rafalskiy** ⓘ *, **Ivan Mershiev, Ekaterina Moiseeva**

Immanuel Kant Baltic Federal University, Kaliningrad, Russia

* V.Rafalskiy@mail.ru

## Abstract

Cardiovascular disease is the leading cause of morbidity and mortality worldwide. Long-term use of antiplatelet drugs is a well-studied therapy for the prevention of cardiovascular death. Ensuring compliance with lifelong administration of antiplatelet drugs, in particular acetylsalicylic acid, is one of the challenges of such therapy. The aim of this study is to explore the possibility of using nuclear magnetic resonance spectroscopy to identify acetyl-salicylic acid metabolites in urine and to search for characteristic markers that could be used to detect patient compliance with long-term acetylsalicylic acid treatment.

## Introduction

Cardiovascular disease (CVD) is the leading cause of morbidity and mortality among individuals worldwide. In 2017, CVD caused an estimated 17.8 million deaths globally, corresponding to 330 million years of life lost and another 35.6 million years lived with disability [1, 2]. For high-risk patients it is important that treatment is continuous and that good compliance is maintained over the long term [3].

Currently, several approaches have been developed and applied in assessing adherence to aspirin: pill counting, interviewing patients, evaluating platelet function, determining aspirin metabolites in blood and urine. Only few studies have measured levels of acetylsalicylic acid (ASA) metabolites in patients with CVD taking ASA for CVD prevention [4, 5].

The biological half-life of ASA is only about 20 min [6], and ASA undergoes rapid hydrolysis by esterase activity in the gastric mucosa, liver, plasma and erythrocytes, producing major ASA metabolites–salicylic acid (SA) and salicyluric acid (SU) as far as minor metabolites–gentisic acid, salicyl-acyl glucuronide, salicylphenolic glucuronide, salicyluric acid phenolic glucuronide and gentisuric acid [7–9]. The urinary excretion ratio of the three main metabolites of ASA in human urine–SA, SU and gentisic acid is about 8.3: 45.0: 1.0 [9]. Therefore, the most rational approach for monitoring ASA ingestion is to identify two main ASA metabolites (SA and SU) in blood or in urine.

Various analytical methods have been developed to determine ASA metabolites in human blood and urine [9, 10], such as high-performance liquid chromatography (HPLC), a combination of liquid chromatography with mass spectrometry (LC-MS and (LC-MS / MS) [9–13],

**Funding:** The studies reported in this publication were supported by a grant from the Russian

Foundation of Fundamental research and government of Kaliningrad region, project No. 19-415-390001.

**Competing interests:** NO authors have competing interests.

gas chromatography in combination with mass spectrometry [14], and ultra high performance chromatography combined with UV or fluorescence detection (UPLC) [10, 15].

Despite the fact that LC-MS / MS and HPLC methods are becoming more and more popular, in recent years there has been a steady increase in the number of metabolomics studies based on the nuclear magnetic resonance (NMR) method, in particular, the study of aspirin metabolites in biological fluids [16].

NMR spectroscopy is recognized as a rapid, non-invasive, reliable method for metabolomic studies in biological fluids, such as urine or blood, and used in order to develop new analytical methods for diagnosing diseases [17, 18]. The main advantages of NMR are the non-destructive nature of NMR spectroscopy and the relative simplicity of sample preparation, as the latter does not require chromatographic separation or chemical derivatization, and high level of experimental reproducibility.

Data extracted from the $^1$H NMR spectrum, such as chemical shifts of various protons, signal multiplicity, line widths and intensities, contain all information about the structure of the compounds under study, which allows them to be identified. Unknown metabolic products can be determined as well.

The advantage of the NMR method over LC-MS is the possibility to quantify more than 200 components of various chemical nature within a single experiment in a relatively short time, within minutes or seconds after the spectra have been collected [18, 19].

However, the NMR method has a number of drawbacks. Compared to LC-MS / MS and HPLC, NMR spectroscopy has low sensitivity. While metabolites with nanomolar concentrations (10–100 nM) can be easily detected using LC–MS, the sensitivity of modern NMR methods is usually 10–100 times less [16].

On the other hand, modern NMR spectrometers with 400–600 MHz frequency allow the detection and quantitative determination of compounds in concentrations of 10–8 M and below relatively fast (about 10 minutes) [17].

The NMR method has been successfully used to detect metabolites in urine and blood in the case of salicylate poisoning and drug overdose [18–20]. However, there is no research focused on ASA analysis of metabolites related to patient compliance monitoring. The aim of this study was to study the possibilities of using NMR spectroscopy to identify ASA metabolites for monitoring patients for compliance with ASA.

In this work we focused our attention on the study of ASA metabolites in urine, taking into account the fact that, in blood, SA produced after administration of aspirin has a biological half-life of 2–4 hours, so the concentration of salicylate in urine, at least in the form its major metabolite SU, is significantly higher than in serum [12, 21] and blood plasma. In addition, urine does not contain the protein component that can bind and mask aspirin metabolites, leaving a small number of unbound metabolites, which can complicate the correct interpretation of the spectrum. Moreover, the chemical structure of ASA and its metabolites ensures that their signals arise in a relatively uncrowded spectrum region of aromatic compounds.

Thus, the main goal of this study was to explore the possibility of using $^1$H NMR spectroscopy to identify ASA metabolites in urine and to search for characteristic markers that could be used to monitor patients for ASA compliance.

## Materials and methods

### Human subject research

The investigation used a panel of healthy volunteers, drawn from the students and staff of Immanuel Kant Baltic Federal University. All subjects were aged between 19 and 50 years old with body mass ranging from 55 to 85 kg. The subjects had taken no drugs for at

least seven days prior to their participation in the study; no other exclusion criteria were applied. Urine samples were collected from healthy volunteers after signing informed consent. All research documents–signed informed consent forms, research protocol, and registration cards were reviewed and approved at the meeting of the Independent Ethics Committee at the Immanuel Kant Baltic Federal University on May 16, 2019 (Committee Minutes No.8).

Urine samples of 4 volunteers were collected before and after the administration of 100, 300 and 3000 mg of ASA. Urine was collected at several intervals within 24 hours after the drug administration. Freshly collected urine was centrifuged for 10 minutes at 7000 rpm (2000g) in order to remove any solid particles, after which the supernatant was collected and passed through a sterile syringe filter with an average pore diameter of 0.2 μm and membrane diameter of 28 mm (cellulose acetate, fiberglass (Merck, Millipore)). Then, 100–200 mM phosphorus buffered solution ($K_2HPO_4$) was added to the urine sample to avoid changes in chemical shifts and to stabilize solution for several hours. The urine pH was adjusted to 7.2–7.4.

Sample volume and pH were recorded during collection, and then samples were frozen and stored at -20 ˚C until analysis. For a more detailed analysis of the kinetics of metabolism, urine samples collected 1.5, 4, 6 and 9 hours after the administration of aspirin were also examined. The samples remained stable and gave repeatable results after 3–4 thawing and freezing cycles.

## $^1$H NMR study of urine samples

For $^1$H NMR experiments, 540ul of prepared and pH-corrected urine sample were placed in a standard 5mm sample tube, then 60ul of $D_2O$ were added to the sample as an internal lock to stabilize resonance conditions. Urine samples were examined by $^1$H NMR on a VARIAN 400 spectrometer in a 9T magnetic field. The main parameters of one-dimensional experiments were as follows: the duration of 90˚ pulses was 2.5 μs, the number of scans varied from 128 to 8192. All spectra were acquired at 25˚ C. The assignment of signals is given relative to TMS. Spectral assignment of endogenous compounds in urine samples was performed by taking into account the literature data on chemical shifts in biological fluids and spin-spin interaction constants [17, 18]. Preliminarily, in order to distinguish the signals of aspirin metabolites against the background of healthy human metabolites, such as hippuric acid, $^1$H NMR spectra of standard compounds, purchased from Sigma-Aldrich (salicylic, salicyluric and gentisic acids, corresponding to the major metabolites of acetylsalicylic acid and hippuric acid) were recorded under the same experimental conditions as urine samples. Additionally, two-dimensional COSY and homonuclear J-resolved (HOMO2DJ) experiments were performed in order to resolve overlapping spectral lines of aromatic protons.

Various water suppression pulse sequences, such as presaturation, WATERGATE, and WET were tested to minimize residual water signal and to increase sensitivity of weak signal detection. WATERGATE technique turned out to be the most effective for recording one-dimensional spectra of the studied samples. It allows obtaining spectra with minimal artifacts in the suppressed water signal region. The WATERGATE is a spin echo sequence using frequency-selective excitation pulses with a minimum excitation power at the frequency of the water signal. The sequence parameters were fine-tuned based on the efficiency of the water signal suppression. The main advantage of the WATERGATE sequence was that it allowed us to observe signals from protons whose frequencies are close to or overlap with the resonance of water (DHO). In two-dimensional COSY experiments, the water signal was suppressed by the WET method.

## Results

$^1$H NMR urine spectra from healthy volunteers, collected in December, November, March and July, prepared as described above, were analyzed to determine seasonal metabolic changes. The most intense signals associated with the known major endogenous compounds: creatinine, urea, TMAO (trimethylamine oxide), glycine (Gl), lactate, citrate, acetate, hippurate, etc. were observed in the 1H NMR spectra. The analysis showed that the profiles of individual spectra practically did not change in the winter, autumn and spring months. However, in urine samples collected in July, there was some change in the 7.0–7.4 ppm aromatic region of the spectrum.

Urine samples from healthy volunteers before and after the administration of 100 mg, 300 mg and 3000 mg of ASA prepared as described above were analyzed by $^1$H NMR using water suppression method. The most significant changes in $^1$H NMR spectra were observed in urine samples from volunteers after the administration of 3000 mg of ASA. We were able to clearly see the emergence of intense overlapping signals in the region between 6.7 and 6.8 ppm, as well as clearly distinguishable triplet and overlapping signals in the regions between 7.2 and 7.4 ppm and between 7.5 and 7.6 ppm. As an example, Fig 1 shows two fragments of the $^1$H NMR urine spectra collected prior to and 2.5 hours after the oral administration of 3000 mg aspirin. NMR spectra are aligned together to better visualize changes. In addition, there is some change in the spectrum profile in the region from 3.5 to 4 ppm. The intensity of the hippurate doublet at 3.8 ppm is significantly reduced, and a singlet appears due to presence of $CH_2$ group of SU in urine samples from volunteers after administration 3000 mg of ASA.

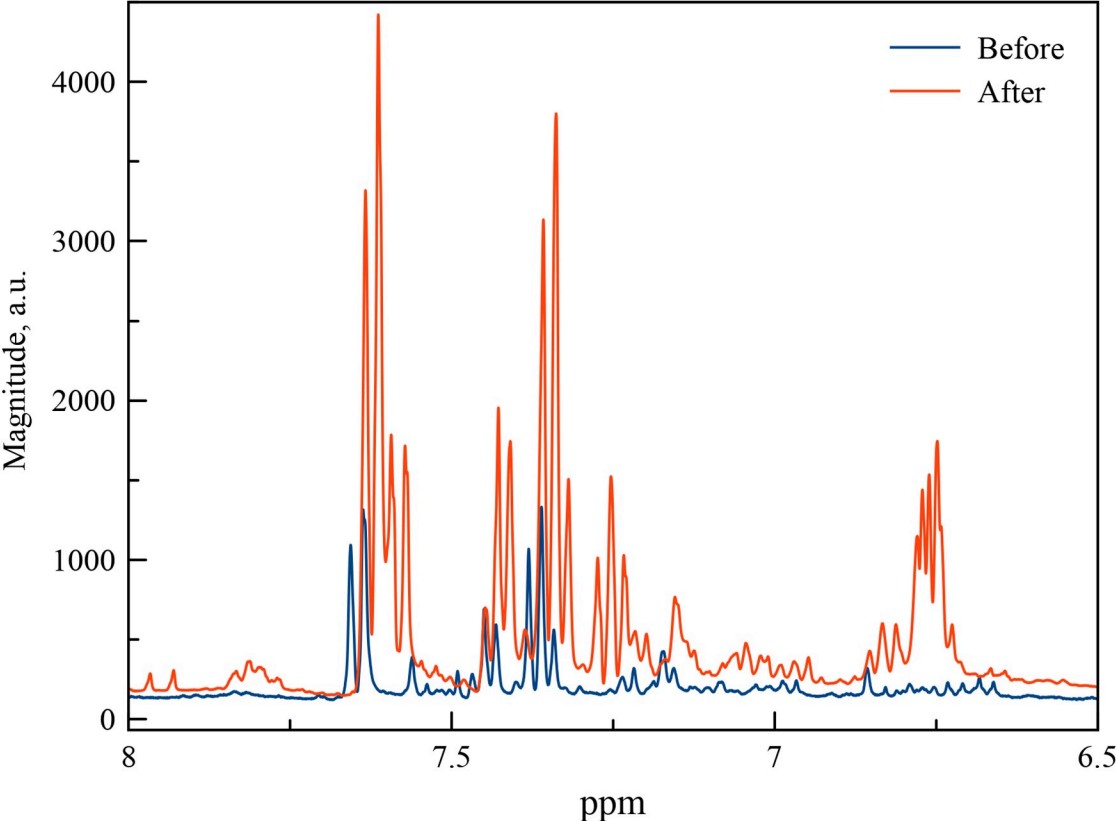

**Fig 1. $^1$H NMR fragments of the spectrum of a volunteer urine sample collected before (blue) and 2 hours after the administration of 3000 mg of the ACA (red).**

It can be assumed that the emergence of new signals is associated with the major metabolites of the ASA such as SA and SU. The integrals of the arising peaks at 7.6 (double doublet), 7.25 (double triplet) and 6.7 ppm (overlapping signals) exhibit classic ratio of 1:1:2. This indicates that the peaks observed at 7.6 ppm and 7.25 ppm correspond to one H atom. A group of overlapping signals at 6.74–6.77 ppm region corresponds to two protons, if we assume that the doublet and triplet are due to the remaining two H atoms on the aromatic ring. To identify signals, the spectra of urine samples were compared to the spectra of pure hippuric acid, SA and SU, which were acquired earlier.

To unambiguously assign overlapping signals and identify metabolites, $^1$H 2D NMR spectra of COSY and HOMO2J were acquired. The COSY spectra of urine obtained before (red) and after (green) the administration of 3000 g ASA are shown in Fig 2. Cross-peaks in the 7.3–7.7 ppm region caused by the interaction of aromatic protons of the hippuric acid. It can be seen that these cross-peaks remain unchanged in spectrum of the urine sample collected after the administration of ASA. The appeared cross peaks at 7.6, 7.25, 6.76, and 6.75 ppm correspond to the interacting aromatic protons of H6, H4, H5 and H3 of the benzene ring respectively.

The HOMO2J spectrum data provided additional confirmation that only the signals of two protons contribute to the overlapping signal at 6.74–6.77 ppm and allowed to determine the spin-spin coupling constants ($J_{6-5,\ 5-4} = 8$Hz, $J_{6-4} = 2$Hz, $J_{4-3} = 7,8$Hz). We concluded that intensity signals arising in the $^1$H NMR spectrum corresponded to SU.

Similar changes in the $^1$H NMR spectra were observed in the urine samples of volunteers after the administration of 100 mg and 300 mg of ASA. We found that the intensities of the signals depended on the time passed between the urine collection and the drug administration, on the one hand, and on the dosage, on the other.

$^1$H NMR fragments of the spectra of volunteer urine samples collected after the administration of 100mg and 300 mg of ASA are shown in Figs 3 and 4. Fragments of HOMO2J spectrum of urine samples collected after taking 300 mg ASA are shown in Fig 5.

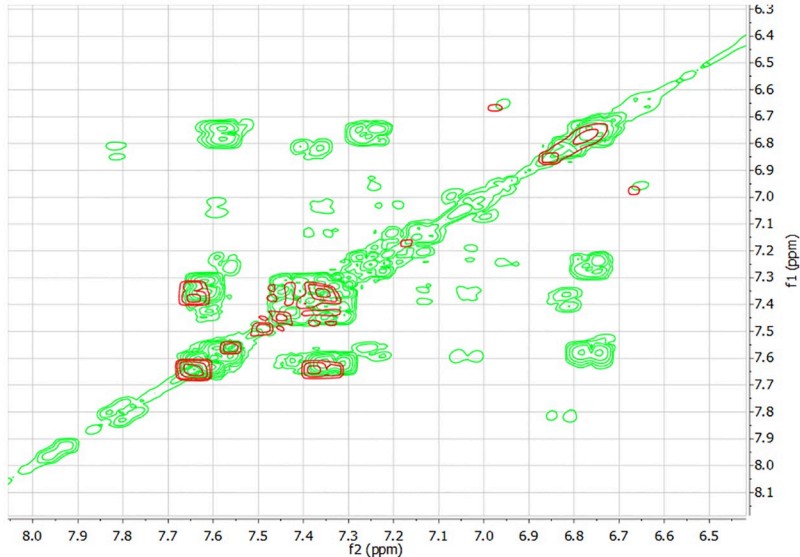

**Fig 2. Overlay of COSY spectra of urine before (red) and post-dose (green) after the administration of 3000mg of ASA.**

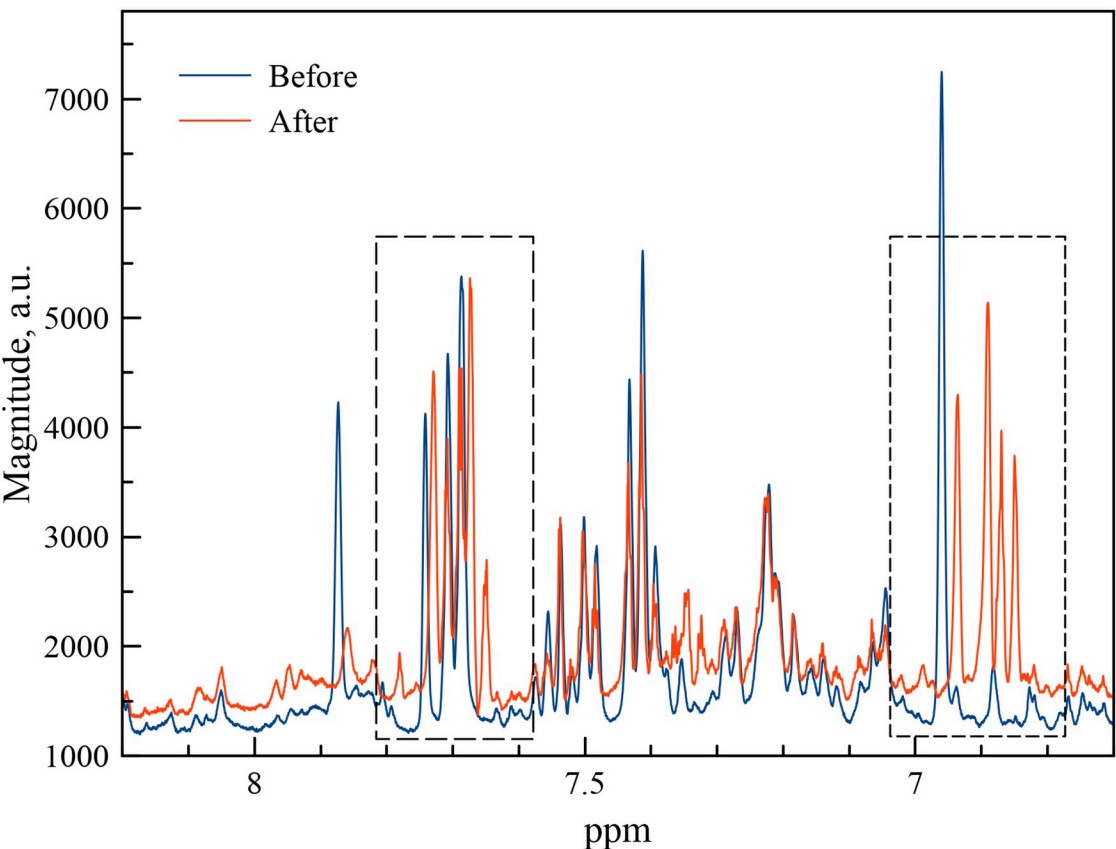

**Fig 3. [1]H NMR spectra of urine samples from a healthy volunteer, collected before (blue) and 4 hours after the administration of 100mg ASA (red).**

Analysis of the 1H NMR spectra of urine samples of healthy volunteers before and after taking aspirin demonstrated different dynamics of metabolism depending on the time of urine collection. There were no significant changes in the spectra of urine samples collected 1.5 hours after taking medication. A slight line shift could be caused by a change in the sample pH. However, the [1]H NMR profile of the spectrum of urine collected 4 hours after administration of 100 mg of the drug shows significant changes. In Fig 3 it can be seen that the line intensities in the 7.3 to 7.7 ppm region change, and new signals appear in the 6.8–7.1 ppm region. More pronounced changes in same regions of the [1]H NMR spectra can be observed for urine samples collected 4 hours and 6 hours after the administration of a single-dose of 300 mg aspirin (see Fig 5).

It is necessary to pinpoint the dynamics of changes in the hippurate signal and the $CH_2$-group signal of SU in the region from 3 to 4 ppm.. The dynamics of the process depends on both the individual characteristics of the volunteers and length of time passed after the drug has been administered. For example, we found that the hippurate signal decreased by 12.3 times 4 hours post-dose for volunteer No. 1, and by 1.77 times for volunteer No. 3. For volunteer No. 2, it decreased by 2 times 3 hours post-dose. The fragments of the [1]H NMR spectra in the region between 2.5 ppm and 4 ppm demonstrate a decrease in the hippurate signal 4 hours post-dose, with the signal's recovery 9 hours after 300 mg of ASA were administered (see Fig 6). Signal changes were calculated in relation to the signal of $CH_2$ group of creatinine.

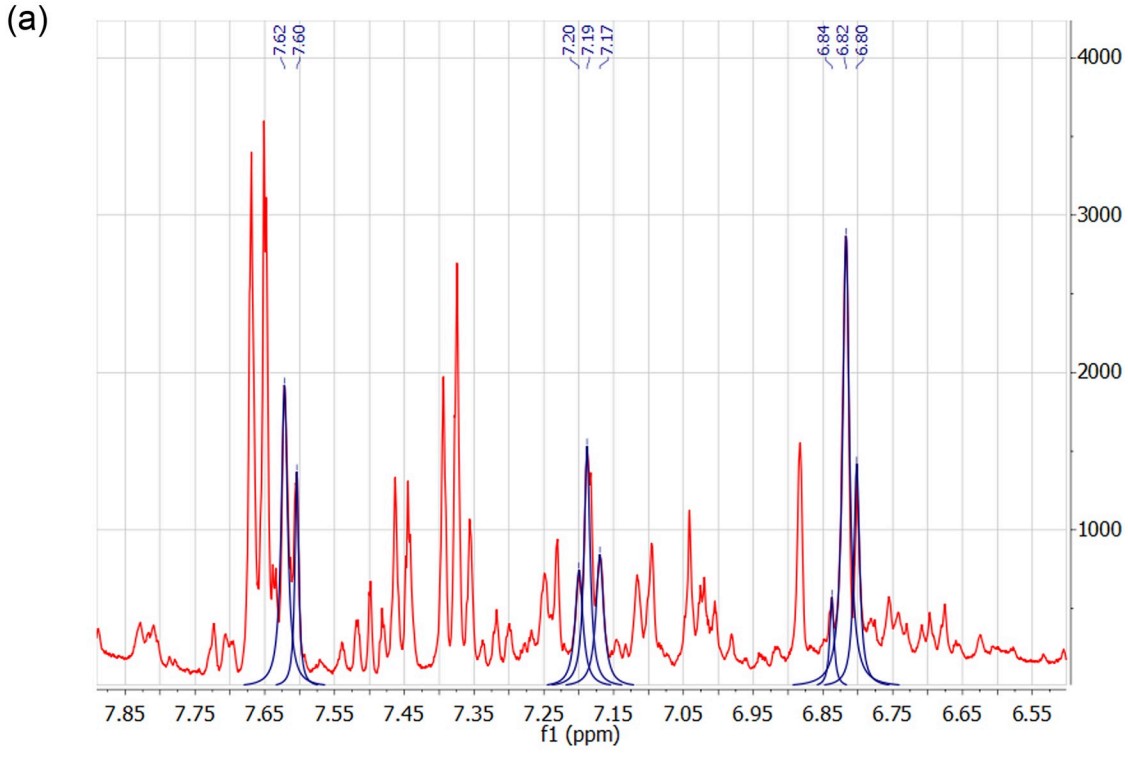

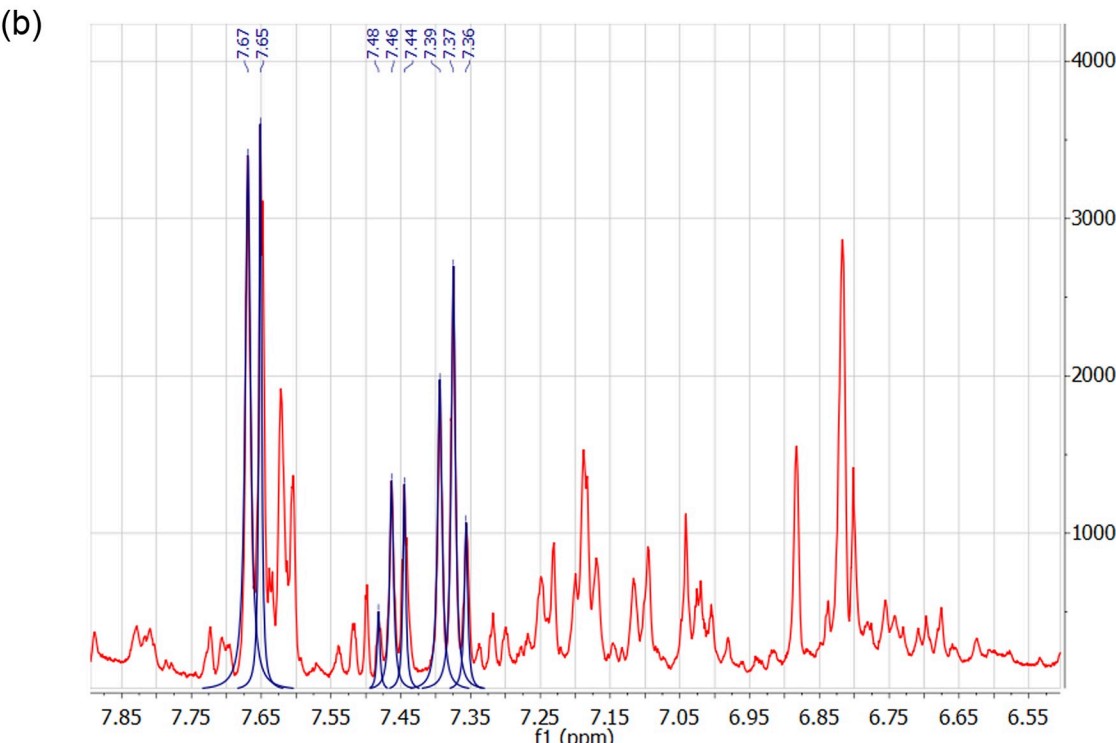

**Fig 4. $^{1}$H NMR fragment of the spectrum of a urine sample collected after the administration of 100 mg of the drug.** The spectrum is compared to the spectra of salicylic and salicyluric acid (a) and with the spectrum of hippuric acid (b).

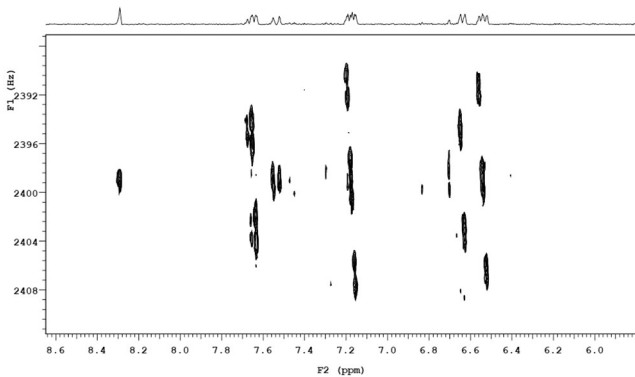

**Fig 5. The fragment of HOMO2D spectrum in the region from 8.5 to 6.0 ppm of urine sample from participant after the administration of 300mg of ASA.**

## Discussion

The study of urine samples of volunteers before and after oral administration of 100, 300, and 3000 mg of aspirin revealed significant changes in one and two-dimensional [1]H NMR spectra associated with the presence of additional signals attributed to salicyluric acid. A review of the literature reveals the variability of data on the content of the major metabolites of ASA in urine after taking aspirin [18–20]. The relative concentration depends on the dose taken by subjects and on the kinetics of excretion of metabolites.

One study presented a case of salicylate poisoning in a 15-year-old girl after taking about 18 g of lysine acetylsalicylate [18]. An urine sample collected after about 6 hours post-dose was analyzed by [1]H NMR. Three types of major ASA metabolites were found, such as SA, SU and GU. The quantitative assessment was performed using the integration of the peaks of the [1]H spectrum. The result showed that the concentration ratio of metabolites SA, SU and GA was 75.4: 20.2: 4.4, respectively. However, only SU was diagnosed by the classical Trinder method, and the SU concentration was approximately equal to the sum of the SA and SU concentrations determined by [1]H NMR.

It should be noted that the data on the concentrations of ASA metabolites vary greatly depending on the study method [18–20]. Three major metabolites were also identified in urine [1]H NMR spectra from drug overdose cases, while gas chromatography and mass spectroscopy studies of these samples did not reveal any organic acids [19, 20].

Literature also presents some data on the time dependence of the decrease in the proportion of ASA metabolites [19, 20]. Thus, one study looked at the effect of the excretion rate of ASA metabolites in pregnant women [22]. It was found that peak detection of SU in the participant occurred at 3 hours post-dose (after ingestion of 75 g aspirin), and at 6 hours post-dose the signal decreased 4 times, having reached the initial level after a day. However, for several participants of the study SU remained detectable above baseline for up to 10 hours post-dose [20].

The effect of therapeutic doses of aspirin on the NMR profile of urine was studied for two healthy participants in [1], albeit with no indication of the dosage used. Still, the researchers concluded that SU concentration in urine showed a rapid increase during the first 7–8 hours, reached its peak in the first 20 hours, and then slowly decreased until completely disappearing 28 hours post-dose. The same study detected no SA signal [1].

The issue of the individual metabolic rate of ASA in healthy volunteers is still unresolved. The naturally-occurring levels of salicyrulic acid (SU) and salicylic acid (SA) excreted daily in the urine of non-salicylate non-vegetarians and vegetarians and patients taking low doses of

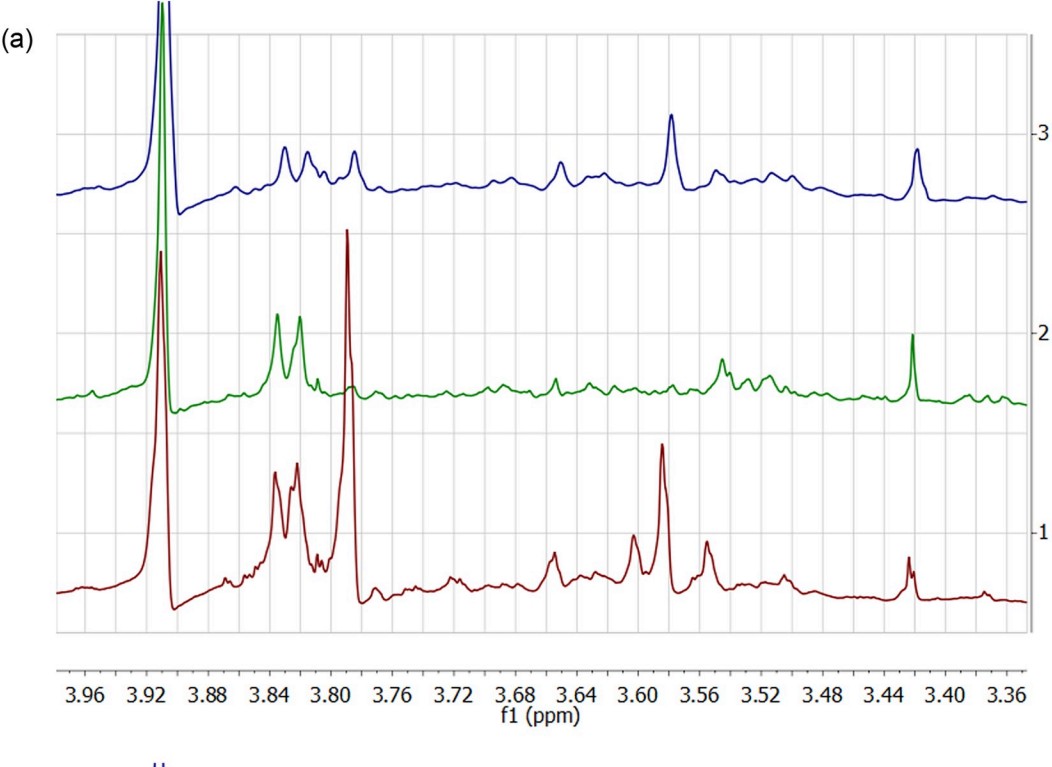

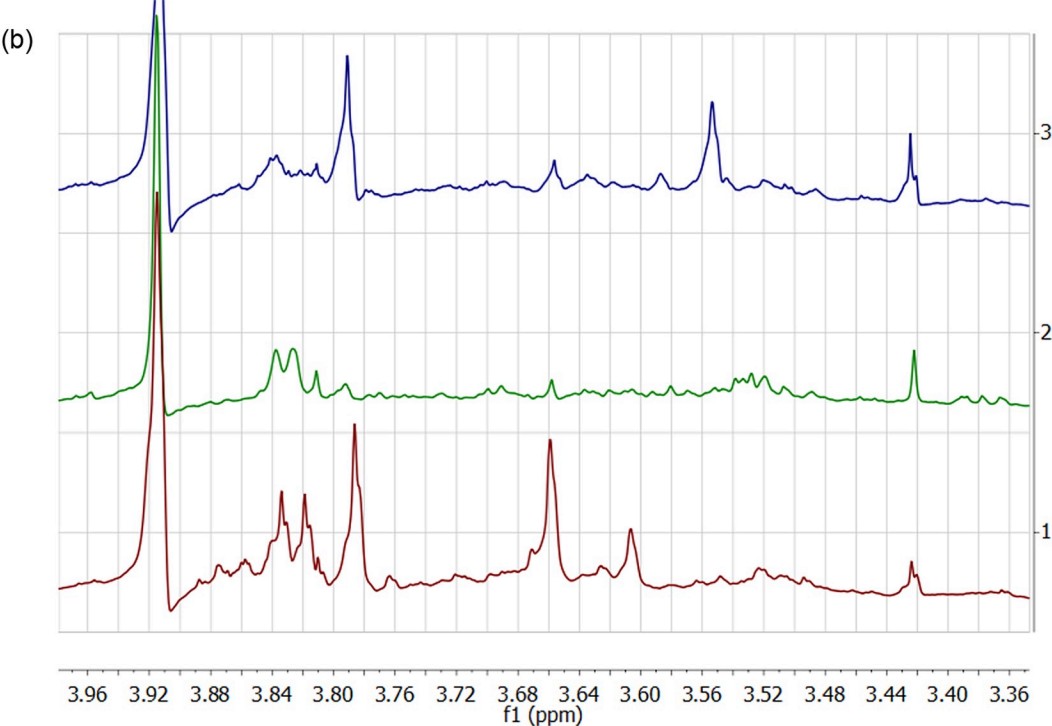

**Fig 6. Fragments of $^{1}$H NMR spectra of urine samples collected before (a) and after the administration of 100 mg of aspirin (b).** Urine samples were collected 4 hours post-dose for volunteers No.1 and No.2, and 3 hours for volunteer No.2.

aspirin (75 or 150 mg aspirin / day) were studied in [21]. The authors found that, daily, vegetarians excreted significantly more SU (4.98–26.60 μmol / 24 hours) than non-vegetarians (0.87–12.23 μmol / 24) hours), although in both cases the amounts were much lower than those excreted by patients taking aspirin. This needs to be considered for identifying low-dose aspirin use. An additional effect that was observed in the NMR spectra associated with a decrease in the peaks of urea and hippurate in urine samples of volunteers after taking the drug must also be noted.

## Conclusion

In this work, we found that in healthy volunteers who took therapeutic doses of 100, 300 and 3000 mg of aspirin, salicyluric acid accounts for the most intense peaks of aspirin metabolites in $^1$H NMR spectra, with the most intense group of signals observed in the region between 6.6 and 6.9 ppm. This group of signals does not overlap with the signals of natural metabolites or with the signals of hippuric acid. This can be one of the indicators of aspirin intake and may be used to monitor ASA in dosages used for the prevention of cardiovascular diseases.

Additionally, we observed the effect of reducing the signals of urea and hippuric acid in the range from 4 ppm to 3.5 ppm. However, a number of issues remain in the development of markers and identification signs based on the NMR method. These issues are primarily associated with individual characteristics of a patient's metabolism. There are still unresolved questions related to the choice of the optimal time for urine collection for monitoring purposes in patients of different health groups who take aspirin in therapeutic doses. According to our study, the optimal SU peaks were observed at 4–6 hours post-dose in healthy participants and the peaks disappeared after 24 hours. However, the data presented in other literature differ markedly from our results [18, 20].

This study showed that the $^1$H NMR is a fast, effective, and non-destructive method for the detection of aspirin metabolites. It does not require laborious sample preparation and can serve as a basis for the development of algorithms for automatic monitoring of aspirin signatures in urine.

## Author Contributions

**Conceptualization:** Galina Kupriyanova.

**Data curation:** Galina Kupriyanova, Ivan Mershiev, Ekaterina Moiseeva.

**Formal analysis:** Galina Kupriyanova, Ivan Mershiev.

**Funding acquisition:** Galina Kupriyanova.

**Investigation:** Galina Kupriyanova, Vladimir Rafalskiy, Ivan Mershiev, Ekaterina Moiseeva.

**Methodology:** Galina Kupriyanova, Ivan Mershiev, Ekaterina Moiseeva.

**Project administration:** Vladimir Rafalskiy.

**Supervision:** Vladimir Rafalskiy.

**Validation:** Galina Kupriyanova, Ivan Mershiev.

**Visualization:** Galina Kupriyanova, Ivan Mershiev.

**Writing – original draft:** Galina Kupriyanova, Ivan Mershiev.

**Writing – review & editing:** Galina Kupriyanova.

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
