## [Decision Letter · Decision Letter 0]

15 Jun 2020

PONE-D-20-12512

Detection of acetylsalicylic acid metabolites in human urine for monitoring adherence to aspirin: the potential of NMR and SERS spectroscopy

PLOS ONE

Dear Dr. Rafalskiy,

Thank you for submitting your manuscript to PLOS ONE. After careful consideration, we feel that it has merit but does not fully meet PLOS ONE’s publication criteria as it currently stands. Therefore, we invite you to submit a revised version of the manuscript that addresses the points raised during the review process.

Specifically, both reviewers raised a number of concerns that should be addressed before the manuscript becomes acceptable.

We look forward to receiving your revised manuscript.

Kind regards,

Oscar Millet

Academic Editor

PLOS ONE

Journal Requirements:

2. Please amend the manuscript submission data (via Edit Submission) to include authors Galina Kupriyanova,  Andrey Zyubin, Ivan Mershiev, Karina Matveeva,Ekaterina Moiseeva, Elizaveta Demishkevich and Ilia Samusev.

3. Please ensure that you refer to Figures 5, 6 and 7 in your text as, if accepted, production will need this reference to link the reader to the figure.

Additional Editor Comments (if provided):

Reviewers' comments:

Reviewer's Responses to Questions

**Comments to the Author**

1. Is the manuscript technically sound, and do the data support the conclusions?

Reviewer #1: No

Reviewer #2: Yes

2. Has the statistical analysis been performed appropriately and rigorously? 

Reviewer #1: No

Reviewer #2: N/A

3. Have the authors made all data underlying the findings in their manuscript fully available?

Reviewer #1: Yes

Reviewer #2: Yes

4. Is the manuscript presented in an intelligible fashion and written in standard English?

Reviewer #1: No

Reviewer #2: Yes

5. Review Comments to the Author

Reviewer #1: The authors failed to prove applicability of the methods (SERS and IR in particularly) for the determination of ASA metabolites in urine. In more detail:

- The paper seems more like a compilation of poorly performed single experiments than the comprehensive study. 

- The analytical part is very poor and does not enable to conclude whether SERS (and other methods) are really capable for precise and accurate quantitative determination of the metabolites in urine or not. The authors write: «The NMR method was sensitive to the detection of low concentrations of ASA metabolites in the urine (at a dosage of 100 mg)». However, quantitative information about concentrations of the metabolites in urine absents within whole paper, i.e., the relationship between metabolite concentration in urine and dosage (or, more importantly, therapeutical effect) has not been demonstrated and proved. The reproducibility and accuracy of the results were not investigated as well. This is a significant disadvantage if to account highly deviating composition of the urine and, consequently, its deviating background signal.

- The results in Fig. 7 and 8 are failed to confirm the presence of SA in the urine samples because the spectra of contaminated samples are too different compared to the spectrum of SA standard.

- The paper lacks an adequate interference study. Although the authors know that urine is a complex mixture, they estimated only contribution of urea to the signal. However, there are numerous other analytes which can contribute to the final SERS signal, e.g., creatinine, uric acid, various metabolites of hemoglobin (e.g., urobilin), etc. All these components possess large Raman cross-section and can generate significant background signal making determination of ASA metabolites impossible. 

General comments:

0) The authors should clearly formulate why someone need to determine SA and SU in urine.

1) The level of English is sometimes quite poor. Also, there are numerous typos.

2) The advantages and limitation of the methods compared to other well-known methods should be analyzed and discussed.

3) The authors should select only one of the used methods and perform really comprehensive study collecting appropriate value of statistical data under supervision of qualified analytical chemist.

Reviewer #2: Reviewing the manuscript entitled “Detection of acetylsalicylic acid metabolites in human urine for monitoring adherence to aspirin: the potential of NMR and SERS spectroscopy”. The authors employed Surface-Enhanced Raman Spectroscopy (SERS), IR and different NMR-Based approaches as a proposed tools to study the cardiovascular patient compliance for long-term treatment of acetylsalicylic acid (ASA). The results showed that both NMR and SERS are powerful approaches and more efficient that IR in detecting the main metabolites of ASA in the studied urine samples. The presented experiments are well designed and the reported results and the conclusion were discussed elegantly, that make it acceptable for possible publication. However, the noted comments pointed at the attached PDF file should be addressed in order to improve the manuscript quality before publication.

6. PLOS authors have the option to publish the peer review history of their article (what does this mean?). If published, this will include your full peer review and any attached files.

Reviewer #1: No

Reviewer #2: Yes: Abdul-Hamid Emwas

---

## [Author Response · Author response to Decision Letter 0]

22 Jan 2021

the authors have completely revised the submitted article

---

## [Editor Report · Decision Letter 1]

2 Feb 2021

NMR spectroscopy reveals acetylsalicylic acid metabolites in the human urine for drug compliance monitoring

PONE-D-20-12512R1

Dear Dr. Rafalskiy,

We’re pleased to inform you that your manuscript has been judged scientifically suitable for publication and will be formally accepted for publication once it meets all outstanding technical requirements.

Kind regards,

Oscar Millet

Academic Editor

PLOS ONE
---

## [Editor Report · Acceptance letter]

23 Feb 2021

PONE-D-20-12512R1 

NMR spectroscopy reveals acetylsalicylic acid metabolites in the human urine for drug compliance monitoring 

Dear Dr. Rafalskiy:

I'm pleased to inform you that your manuscript has been deemed suitable for publication in PLOS ONE. Congratulations! Your manuscript is now with our production department. 

Kind regards, 

on behalf of

Dr. Oscar Millet 

Academic Editor

PLOS ONE